# HIGEA: An Intelligent Conversational Agent to Detect Caregiver Burden

**DOI:** 10.3390/ijerph192316019

**Published:** 2022-11-30

**Authors:** Eugenia Castilla, Juan José Escobar, Claudia Villalonga, Oresti Banos

**Affiliations:** 1Department of Computer Engineering, Automation, and Robotics, University of Granada, 18014 Granada, Spain; 2Department of Software Engineering, University of Granada, 18014 Granada, Spain

**Keywords:** chatbot, conversational agent, Zarit Test, DialogFlow, Telegram, Firebase

## Abstract

Mental health disorders increasingly affect people worldwide. As a consequence, more families and relatives find themselves acting as caregivers. Most often, these are untrained people who experience loneliness, abandonment, and often develop signs of depression (i.e., caregiver burden syndrome). In this work, we present HIGEA, a digital system based on a conversational agent to help to detect caregiver burden. The conversational agent naturally embeds psychological test questions into informal conversations, which aim at increasing the adherence of use and avoiding user bias. A proof-of-concept is developed based on the popular Zarit Test, which is widely used to assess caregiver burden. Preliminary results show the system is useful and effective.

## 1. Introduction

According to the World Health Organization [1], one out of four people in the world will be affected by mental or neurological disorders at some point in their lives, and 450 million people currently suffer from such conditions. As reported by the Alzheimer’s Association [2] in their 2020 Alzheimer’s Disease Facts and Figures Report, more than 5 million Americans are living with Alzheimer’s. By 2050, this number is expected to rise to nearly 14 million. One in three seniors die with Alzheimer’s or another dementia. These diseases kill more people than breast cancer and prostate cancer combined. There is no question that finding a cure to these disorders is a worldwide mission, but as the task is taking place, the carers of patients that suffer these illnesses are often unattended.

From a young age, we hear about these mental disorders. However, we often know little of these disorders and their implications unless we get the chance to take care of a patient ourselves. Not only are mental health disorders one of the fastest growing conditions but they are also the ones that receive most of the caring from family members at some point. These are family members that are most frequently not properly trained. According to [2], more than 16 million Americans provide unpaid care for people with Alzheimer’s or other dementia. These caregivers provide an estimated 16.6 billion hours of care valued at nearly 244 billion dollars. More shockingly, 83% of the help provided to older adults in the U.S. comes from family members, and nearly half of all caregivers who provide help to older adults do so for someone living with Alzheimer’s or another dementia.

As it could be expected, family members that have to care for their relatives often suffer themselves symptoms of depression, anxiety, etc., which are originated not only because of the pain of having to see a loved one go but also because of the burden and expectations that come with the caring task. Fortunately, caregivers are receiving more attention from relevant institutions, as it has been proven that depressive symptoms among patients are associated with the negative mental state of caregivers [3].

Over the years, various tests have been developed to assess the level of burden of the caregivers. These tests are often carried out sporadically, and they consist of very personal yet revealing questions that help to understand how the caregiver feels. One of the best known tests is the Zarit Burden Interview or Zarit Test, an interview that consists of 22 questions reporting on topics such as economic means, personal experiences, and expectations. The caregiver needs to evaluate each statement from a range of 1 to 5, being 1 “never” and 5 “nearly always”. While these and similar tests have been proven effective in evaluating the level of burden in caregivers, they present some practical limitations. For example, they are normally administered by an expert during a doctor or routine visit, most frequently as a consequence of the caregiver showing already symptoms of burnout or depression, thus hindering its preventive diagnosis capacity and also providing a limited picture of the caregiver burden evolution. In addition, the tests include several questions to go through in a single shot, questions that tend to be quite straightforward and sometimes hard to answer, as they present the caregiver with a painful reality. This can result in worst-case scenarios in a worsening of the caregiver mental state. Moreover, these tests are normally filled in a hand-written fashion, thus potentially leading to data loss.

In the light of the existing limitations of classical ‘pen and paper’ evaluation tests, and in view of the potential offered by current digital technologies, we propose a novel system aimed at facilitating the ubiquitous and convenient realization of caregiver assessment tests. Namely, in this paper, we present HIGEA, a mobile-cloud-based system based on a conversational agent that naturally embeds caregiver burden assessment test questions into informal conversations and over time. In this way, the system intends to reduce the caregiver negative awareness caused by the nature of the questions and lessen the sometimes harmful side effects of having to reflect on their daily reality. In doing so, HIGEA aims to increase the adherence of use and to also avoid user bias due to learning effects. The conversational agent is further aimed at capturing the caregivers responses at different moments in time, thus potentially filling the gaps typically existing in the classical administration of these type of tests. The system is designed to run on any mobile platform regardless of the underlying operating system. A proof-of-concept version of HIGEA is developed based on the popular Zarit Test to showcase the usability and interest of these type of tools.

The reminder of the paper is as follows. Section 2 presents the literature related to the use of conversational agents in mental health applications. Section 3 describes the proposed system and its technical requirements. The evaluation of the system is detailed in Section 4. Finally, the main conclusions of the paper are provided in Section 5.

## 2. Related Work

Conversational agents have been used in some mental health and well-being care problems. For example, Bird et al. [4] provided evidence that web-based interventions using a chatbot interface can help with resolving distress issues or at least reducing its effects, even for those users experiencing high levels of distress at baseline. For their study, they used two well-known chatbots: ELIZA and a self-help computer software package called Manage Your Life Online (MYLO). They were not the first to try MYLO in a survey. Gaffney et al. [5] also conducted a survey to study the efficacy of MYLO. They also concluded that MYLO and ELIZA were associated with reductions in distress. After comparing data from websurveys and chatbots, Kim et al. [6] concluded that conversational agents can be useful for reducing distress. Namely, the authors showed that the use of chatbots encourages the interaction of the user; the answers were of a higher quality and more reliable.

Gillian et al. [7] developed a chatbot intended for mental health counseling. Although the prototype of their conversational agent was simple, their paper concluded that the use of chatbots were rather useful, as the research showed that the users found the chatbot “safe” and easy to use. This proves also that the implementation of chatbots can help patients that feel uncomfortable with a face to face conversation. Siddig et al. [8] designed and implemented PlyBot, which is a psychological self-help chatbot. PlyBot was a rule-based chatbot developed in a cloud-based platform. They found the use of cloud-based platforms very convenient for their work, as they provided many advantages for both users and psychologists. They concluded that this technology enables users to easily obtain access to the chatbot and help psychologists gather more specific, documented, and safe information.

Conversational agents have been limitedly focused on addressing caregivers problems. Nonetheless, there are some works contributing in this direction. For example, Joerin et al. [9] conducted a study in which caregivers of patients with mental health illnesses tried out a chatbot called Tess. This project included the participation of professional counselors, practitioners, caregiver coaches, and the use of specific material to create content that would align with the needs of caregivers. The data gathered by caregivers from the use of Tess are extraordinary. Not only is the chatbot cost-effective but caregivers found the conversation to be helpful roughly 88% of the time, while three-fourths of the participants found the offered support and coping skills to be much more relevant.

Although there are important developments in the area of chatbots, not many conversational agents specifically focusing on caregivers of mental health disorders such as dementia or Alzheimer’s exist to date. Hence, caregivers often find themselves not knowing what to do or where to ask for help. Giving feedback and support requires understanding what helps caregivers the most when coping with stress, depression, or grief. Some researchers have investigated different methods of dealing with these mental disorders. Cooper et al. [10] concluded that the best practices to cope with the burden are emotion-focused strategies, whereas those who use problem-focused strategies tend to be more anxious one year after the intervention. The emotion-focused mechanisms include exercises for acceptance, humor, and positive reframing. Problem-focused strategies include active coping practices or planning. Religion is also a very popular and effective coping mechanism among caregivers according to Stolley et al. [11]. Information about institutionalization is normally not considered, as some studies suggest that there is no relation between the relief of the caregiver and the institutionalization of the patient [12].

## 3. HIGEA: A Conversational Agent for Caregivers

Helping caregivers should be a priority in current times. Nowadays, people responsible for the ones in need of care are alone in a society that still is unaware of mental health problems and even holds prejudices against them. With the hope of normalizing mental health disorders, and with the main goal of helping caregivers, we propose HIGEA. This new system consists of a public cross-platform chatbot that operates in standard smartphones. HIGEA benefits from the potential of the cloud to delegate major processing and reasoning responsibilities. In the following, we describe the requirements, architecture, and implementation of an initial approach to this concept.

### 3.1. Requirements

To prioritize the requirements of this project, the MoSCoW [13] approach has been used. It was decided very early on that the tool had to work on modern mobile devices. Since the principal goal of HIGEA is to have a very easy-to-use app, the possibility of deploying it to various chat-based platforms (such as Telegram or Whatsapp) is crucial. To keep information through time, the system must be able to save data and interpret it. This is much required since all the answers from the Zarit Test questionnaire have to be saved and evaluated. In order to implement the Zarit Test, the chatbot must be able to embed the questionnaire. The agent must also try and help the caregiver as much as possible; thus, the chatbot should provide feedback in the form of coping mechanisms based on their psychological assessment (e.g., useful tips to the user). It is also a must to reproduce some kind of small talking, so the process of evaluating the level of caregiver burden does not become a tedious task for the user. In fact, answering the whole Zarit questionnaire directly as is, all at once, can produce tiredness and make the user lose interest and precision in the answers; hence, it is something to be avoided.

The chatbot should be able to interact with the user periodically. The Zarit Test is normally conducted every now and then, causing a feeling of abandonment to the caregiver. HIGEA should salvage that by being the one to start a conversation with the user if a certain period of time has passed without any interactions.

Ideally, advanced artificial intelligence algorithms could be used, along with a complex Natural Language Processor (NLP) to improve the answers and be more dynamic. It could also be interesting to integrate this chatbot with external applications used by healthcare professionals to help track the quality of life of caregivers and improve the medic–patient relationship.

### 3.2. Architecture

The design choices made for the proposed system are highly influenced by the type of chatbot to be used. We introduced earlier the two main categories of chatbots, RB- and AI-chatbots. Both have pros and cons, and once more thanks to technology, we can have a bit of both worlds. This can be achieved with the implementation of off-the-shelf cloud platforms. Cloud-based chatbots offer some advantages: for example, they automatically integrate security and ethics, they are also cheap to implement, and the system can scale on-demand. Furthermore, they are usually compatible with many platforms, so the deployment is vastly simplified.

In Figure 1, the architecture designed for this project is displayed. The system can be divided into three layers or modules, which are named according to their responsibilities: the Presentation module, the Chatbot module, and the Data Storage module. The Presentation module defines the user–chatbot interface, which is here mainly devised as of text type (although multimedia contents are supported too). The Chatbot module embeds the functionalities in charge of unwrapping the user inputs and generating the corresponding responses. The Data Storage module encapsulates the knowledge bases and facilitates the interaction with the Chatbot module in order to perform the reasoning and decision making needed to elaborate the responses to the user.

The normal operational flow is as follows. Upon request or under their own discretion, the user sends a text input to the message application. This application sends the userID together with the text input to the Chatbot Module. This ID is univocal to the message application, so it can be used as the univocal identifier for each user within the Data Storage module. Then, the data are processed by the NLP component. At this stage, punctuation and empty words such as pronouns are erased, sending a processed plain text to the intent detector. Once the intent is recognized by the intent detector, a list of predefined variables go through the token extractor. The token extractor is in charge of identifying these variables within the processed text. Through the data flow, both the userID and the intent are maintained. When all the necessary data are gathered from the text input, the variable list, the userID, and the intent are forwarded to the decision engine, which eventually decides what kind of answer the user should receive. The different type of answers include essentially three categories: the concluding answers which represent the end to a conversation, answers perceived as small talk or answers related to the Zarit Test questionnaires. The decision engine is responsible for saving data (facts) in the knowledge base hosted by the Data Storage module. This information could be available for health experts to analyze upon request. Once the decision engine has selected an option, the rules that identify the type of answer and the userID are sent to the response generator. The response generator acts according to the rules received and then sends the information (e.g., recommendation) to the user. Whatever the type of response that is sent, it is sent to the message application as plain text for the message application to render such text in the correct form.

### 3.3. Implementation

The use of cloud-based platforms offers important advantages such as easy deployment to different platforms as well as the integration of both NLP and rule-based chatbots. There are many cloud-based platforms to choose from: IBM Watson or Azure Bot Service are just a few. After exploring the different alternatives, it was decided to use DialogFlow, which is a service provided by Google for the development of chatbots. DialogFlow offers many advantages, including easy deployment to other platforms and the internal use of NLP, though limited. IBM Watson was also heavily considered but rejected after comparing both interfaces and services provided within the free plan.

Another aspect that needed to be taken under consideration for the implementation phase was the message application to be used. Dialogflow offers an easy deployment to 14 different platforms. For the first implementation of HIGEA, Telegram was particularly chosen. Telegram is a well-known message app that is both available for free for iOS and Android operating systems, making an internet connection the only requirement needed for users. This instant messaging application is also quite popular, as according to the company reports, the number of monthly active users is 700 million people worldwide as of September 2022.

For production hosting, Cloud Functions for Firebase will be used, as DialogFlow is specifically designed to work combined with Firebase. This was also one of the reasons to choose DialogFlow. Firebase also offers both the API to code all our functions and data storage, including that of real-time databases, simplifying the task while ensuring a good connection between all elements.

The communication flow at the application level is abstractly depicted in Figure 2. The user communicates via Telegram with the chatbot, which is directly connected to DialogFlow. Then, DialogFlow uses Firebase to fulfill intentions and store data. Afterwards, the result evaluation and metrics are calculated and then returned to DialogFlow to show them through the chatbot on Telegram (note the double direction arrows).

As mentioned earlier, the Zarit Burden Interview is a test that measures the level of burden experienced by caregivers of patients with dementia. The revised version of this test consists of 22 questions [14]. Each question is a statement which the caregiver is asked to endorse using a five-point scale, ranging from 1 (Never) to 5 (Nearly Always). Some examples of these questions are: *“Do you feel that your relative asks for more help than they need?”* or *“Do you feel embarrassed over your relative’s behavior?”*. Considering the scenario for the performance of the Zarit Test, it comes as no surprise that the data gathered are somewhat weak. Not only is the test taken just sporadically but also, given its length, the attention paid to the questions diminishes over time, thus affecting the quality of the responses. This process is also limited by the moment to which it is taken. With the use of chatbots, the test can be delivered little at a time, and the final result tends to be more fair. For example, on the first day, the chatbot may ask the user questions 1 to 6. The next day, it may ask questions 7–15, and so on. In such a way, the test can be responded to under different circumstances, and the user will not get too tired. This also helps with the fact that some questions are very hard and personal to answer, and having to face them all at once can affect the mental state of the caregiver.

The interpretation of the final score is fairly easy. If the sum of all the points returns a score in the range of 0–20, the user suffers from little to no burden. If it is in the range of 21–40, a mild to moderate burden is detected. From 41 to 60, the caregiver suffers from moderate to severe burden, and last but not least, scores in the range of 61–88 indicate the severe burden of the caregiver. Depending on the results, the chatbot will answer one way or another, following the guidelines of Cooper et al. [10]. It should also be mentioned that since the complete administration of the test may take a few days, consequently obtaining all scores, meanwhile, the bot can offer motivational quotes as part of the everyday conversation with the caregiver or helpful information along with small talk to make the task more natural and appealing.

For the purpose of organizing the conversation and giving it a little bit of context, the questions from the Zarit Test have been divided into four conversational categories, which are represented in Table 1. Please note that these categories are completely subjective. When a conversation is taking place, the chatbot first asks all the questions of one category, and when it is finished, it randomly selects another category until all questions are responded. The reason for this division of questions into categories is to give a little bit of continuity to the conversation. For example, if the chatbot is asking the user about hobbies or friends, it would make more sense to ask questions about their social life rather than a question regarding money.

The next step is to actually implement the conversation workflow. Since HIGEA’s intention is to give personalized feedback, it needs to identify and get to know the user a little bit. That is why, when a user is first introduced to HIGEA, an introductory chat takes place. Some examples of this type of interaction are shown in Figure 3.

Once the user is registered in the database, it is time to start asking Zarit Test questions. To make the process less repetitive, random information is given between each question as it is shown in Figure 4. It should be kept in mind that the conversation may seem random, but it has been carefully designed so that it always has something to do with the context of the group of Zarit Test items that are being asked (Table 1).

To make the chat a bit more lightweight, multiple elements of visualization are integrated. For example, buttons may appear from time to time, images, cards, etc. This comes in handy when, for example, the user wants to see information about a support group, such as in Figure 4. With the different types of visualization techniques, information can stand out and seem more appealing to the user. The caregiver can also give feedback onto the information received. This can be helpful to HIGEA, as in this way and over time, it may decide to not show the elements that have received negative feedback.

When asking a Zarit Test question, the visualization may differ as well as it is observed in Figure 5. Sometimes, the user will answer with buttons displayed in the chat itself, and others, for example, the keyword may transform itself into buttons.

HIGEA gives feedback to the user depending on the answers to the Zarit Test questions. In situations where the user is feeling uninformed, the chatbot might offer some references to support groups. If the user is feeling stressed or anxious, the agent may send, if requested, information on relaxation techniques. These techniques have been selected after reading diverse articles that back up these proceedings as effective, and the suggestions have been gathered from the most reliable sources possible. For example, the information about economic aid has been gathered from official sites such as *The Bright Focus Foundation* [15], the *Alzheimer’s Association* [2] or the *Dementia Care Central* [16]. It is important to note that by showing these cards, the system is not appropriating the intellectual value of its content, as the cards only show an introduction and a button to go to the website where all the information is, not hiding in any way the authorship of the materials. Information about breathing mechanisms was extracted from Medical News Today [17], a web-based outlet for medical information and news. McMaster University and Mindfulness Melbourne [18] offer some interesting examples of muscle relaxation techniques, among others [19,20]. All the visualization techniques have been extracted from Mindful Minutes [21], which is a blog written by a certified instructor in Yoga and Primordial Sound Meditation. In order to add accessibility to the system, the support groups offered are online-based, such as that of the *Alzheimer’s Association* [2] or even Facebook groups such as Memory People [22], the Dementia Caregivers Support Group [23], or the Alzheimer’s and Dementia Caregivers support chat group [24]. Feeling alone is a common symptom of caregivers [25]; that is why support groups are important for the user. Another method might be to show real testimonies of caregivers around the world. The Family Caregiver Alliance [26] has a great and extensive collection of testimonies.

Some articles point out the importance of including in the conversation some related topics that typically concern caregivers such as salaries [27] or financial burden [28]. Other contents, such as for example the Mindfulness-Based Cognitive Therapy (MBCT) [29,30], breathing mechanisms and yoga [31,32,33], muscle relaxation and visualization techniques [34,35,36,37] are also included due to their efficiency in reducing stress and managing difficult situations better. Some examples of this type of feedback are shown in Figure 6.

Despite the interest of informing caregivers on the aforementioned matters, ensuring the adherence to the use of a system such as HIGEA requires taking a step forward. Foremost, people want to feel heard. Moreover, the conversation cannot be fully spontaneous; it needs to follow a proper line of reasoning. Hence, a very specific logic has to be followed. The diagram shown in Figure 7 showcases this logic. When a registered user, that is a person who has interacted with HIGEA, decides to speak to the chatbot again, HIGEA will choose a topic of conversation. From then on, depending on the user’s answers and some randomness, HIGEA mixes both normal topics of conversation with Zarit Test questions. It should be clarified that the intention of HIGEA is not to fully simulate a human conversation, which to date is out of the reach of most conversational agents, but to facilitate the care burden syndrome assessment. The goal is then to ensure that the Zarit Test questions are properly asked and responded.

Once the user has answered all of these questions, the system calculates the scores and brings the user new feedback tailored to their results. The abundance and originality in the feedback provided to the user is very important so that the caregiver is not reminded that they are talking to a computer program. Moreover, when all the responses are gathered, the process of evaluating the Zarit Test begins. It should be noted that answering all the Zarit Test questions may take days sometimes, depending on how often the caregiver speaks with the conversational agent. The questions do not account for immediate feelings or thoughts but rather longer-term reflections, thus not posing any significant issue response-wise. The evaluation process starts only when all 22 questions have been registered in the database as answered.

According to the Zarit Burden Interview, results can be categorized into four scales (Table 2). Since all the answers to the Zarit Test questions are stored in the database, adding up all the answers and calculating the level of burden according to Table 2 is fairly simple. When the final score is calculated, HIGEA is triggered to generate and present the user with the corresponding feedback. There are different options within each category of burden level, as shown in Table 3. Certainly, not all options are displayed at once, but rather some are chosen based on an strict randomization process.

## 4. Evaluation

HIGEA has been devised to be able to offer some small talk, ask psychological test questions, and act upon the answers of the users. To evaluate HIGEA, a group of ten non-professional caregivers were given a phone with the Telegram app installed. They would then interact with HIGEA for some time and by the end of the session were asked to provide their impressions. The demographics of the participants is provided in Table 4.

To study the usability of the system, the users completed the *System Usability Scale* (SUS) [38]. This test consists of a ten-item questionnaire with five possible answers each, from *strongly disagree* (1) to *strongly agree* (5). The effectiveness of this test has been proven in [39,40]. SUS has become a reference in evaluating usability in the industry as it is very easy to administer to participants. It can be used on small sample sizes with reliable results, and it can effectively differentiate between usable and unusable systems. Equation (Equation 1) has been used to calculate the final score of the SUS test.
(1)$$Score=2.5\cdot \sum_{n=1}^{10}\left(\left(\frac{2\cdot n}{n}\in Z\right)\to \left(5-S_{n}\right)\right)\wedge \left(\left(\frac{2\cdot n+1}{n}\in Z\right)\to \left(S_{n}-1\right)\right)$$

### 4.1. Results

The quantitative results of the usability evaluation are depicted in Figure 8. As it can be observed, the participants rated quite positively the questions referring to their willingness for using the proposed system frequently (Q1), the ease of use (Q3), the promptness in learning to use the system (Q7), and the confidence while using the system (Q9). Moreover, users mostly disagreed about the system being unnecessarily complex (Q2), inconsistent (Q6), cumbersome (Q8), and requiring to learn a lot of things before using it (Q10). The participants were generally on the side of not considering much necessary the help of a technical person to use the system (Q4), and they also found that the various functions in the system were well integrated (Q5). All in all, these responses translate into overall users’ scores above 68 points, which represents the average standardized SUS score [39]. A system with a score over this value is considered to have a good usability level.

In the following, we present some of the feedback gathered during the informal meetings held with the participants after using the system. Thus, for example, User A stated that she found the tool interesting, although she needed some indications as to how the system actually worked. While she found that the tool required more variety in its feedback, she also stated that she *“considered this tool as an opening door to a more sophisticated instrument to both comfort the caregivers and provide the community with interesting data”*. She also said that the system *“took away a portion of the load of anxiety that caring of an ill-loved implies”*. This user spent a total of a month caring for her father, who lives abroad and is in stage 2 of Alzheimer’s. She said about her experience in caring that it took a toll on her and after using the chatbot is actively seeking more help for her parents. The answers of user B were very similar to that of user A. She is not close to her relative yet she spent some time taking care of them. As for her perception of HIGEA, she stated that she wished the conversation was more like a conversation in between people and suggested that the chatbot should give more personalized feedback in future versions. She also admitted to find the system interesting and the information generally helpful.

User C, who lives with his relative, mentioned that the system offered some help and some useful information. As a matter of fact, he did not know, for example, of the existence of support groups of caregivers and said he would look more into that. He agreed with the other two users that the conversation needed more humanization but that *“it was a good starting point to develop more tools like this one”*.

Users D and E are mother and daughter, respectively. They were not able to use the system for long, but they went through the normal conversation a person is supposed to have with the chatbot as well as some live examples. The situation is a little bit different for these two users, since they live full time with their sick relative, who is not elderly. The feedback received from these two participants was quite similar despite their age gap. While it is true that user D found the system a bit more difficult to understand at first, both ended up thinking that it was easy to use. User E, a teenage girl that worries for her father, and knows what is like to live with someone that has mental health issues, said that she found the approach promising, especially for younger users that are more in touch with technology. She also thought that being advised on associations and help groups was very important for people to be able to share their feelings. User D was very touched by the project and the motivation behind it; she has experienced a big sense of abandonment from the public health care system and identified with the caregiver burden syndrome. She not only took interest in the tool itself but in the whole project. She said *“this is a real problem, and people need to talk about it, that is why I find specially interesting that the system is able to give information about support groups”*. She also hoped that the system could help break the stigma of mental health diseases in society. Both participants also concluded that the fact that the chatbot had to be more human-like was extremely important and that there was more work to be accomplished in this direction.

Users F and G found the idea behind the system reassuring and accompanying. User G particularly said that when caring for her husband, she felt very alone and almost questioned her own life at some point. She described the illness of her husband as being so severe that she did not recognize him at the end. “He was a different person”, she said. She also said *“I understand that my children have their own life, their own works, their own families... but you do feel alone”*. Hence, an application such as HIGEA could help coping better with the loneliness. User F said that she also felt some level of guilt for not being able to give more time to her parents, and both F and G found especially interesting the extra information provided by the chatbot on financial aid.

User H thought that the bot could be more developed but that it was a great way to keep track of the caregiver burden state. In addition to that, she found the app easy to use from a technical point of view. Both H and G agreed that the system was a good starting point but that there was a lot of work ahead. They thought that having the system integrated in a healthcare environment should be a must. While both participants valued the usability of the conversational agent, they agreed in that more work should be devoted to making the chatbot more human-like.

### 4.2. Discussion

Even though the results are very similar accross participants, it is no surprise that, for example, user B scored higher in the questions that addressed the matter of the system being easy to use in terms of guidance and instructions. This could be due to the fact that the user is younger and is more in contact with different types of applications. User C scored the lowest in the SUS scale, which might be motivated by the fact that he actually lived with the sick family member and felt the need for more personalized help. Hence, the proposed tool might be found insufficient for the high needs of this participant.

In general, the results were positive and the feedback helpful. The fact that three out of five users agreed that the chatbot needed to be more human-like addresses one of the limitations of cloud-based chatbots. These conversational agents are mainly rule-based and need a lot of interaction with the user to develop a consistent and solid argumentation line. It is fair to say that the users did not complain about the conversation being humdrum and gave positive feedback about all the additional information provided to them during the conversation with the system. The issue about the variety in the conversation could however be fixed adding more options for the chatbot to use when answering a user.

There are a fair number of limitations that come with using cloud-based platforms to develop a conversational agent. First of all, even though the development started within the free billing plan of Firebase, the billing plan had to be updated later on to the Blaze plan, i.e., a “pay as you go” philosophy. Even if this was inexpensive, it may be an inconvenience depending on the number of users. The logs that referenced potential issues on the cloud services used were poorly self explanatory and added a lot of lost time trying to fix issues that sometimes had a simple solution. It would also be interesting to analyze better strategies to introduce small talk into the process, although probably, the answer to this issue is to implement the conversational agent with artificial intelligence. From a practical perspective, although more and more people use Telegram worldwide, it is still behind other major instant messaging services such as Whatsapp. In the authors’ view, Telegram provides a much higher portfolio of options to interact with users. We acknowledge the relatively limited use of Telegram as compared to Whatsapp, but we find the former worth using, as it provides a much better user experience.

## 5. Conclusions

The objective of this project was to design and implement a conversational agent that would be able to administer psychological tests to users as part of a regular conversation. In doing so, the system would also have to assess the quality of life of the caregiver and act accordingly. The results gathered make us conclude that the system was, on average, useful. In terms of achievements and requirements fulfilled, the system has provided a psychological test, saved the data for future studies, acted on the users’ answers, and given feedback to them. It has also been successfully deployed to a user-friendly application (Telegram). The chatbot has also been able to introduce a small-talk conversation to a certain degree, although this aspect could be improved.

HIGEA is devised as a tool to inspire the future development of new apps oriented to mental health, especially targeted at caregivers. To the best of our knowledge, HIGEA is the first system of its kind putting the focus on caregivers beyond patients and practitioners. HIGEA is seen to contribute to not only the future of conversational care agents but also to the integration of these chatbots with external health care applications. This way, health care professionals could have a better understanding of the development of the caregiver burden. Medics could also, through the collected data, give more helpful feedback to caregivers. To that end, future work should consider enriching the collected data via the existing caregiver burden tests with additional and possibly more detailed data on factors that may potentially amplify or reduce the perception of burden. For example, wearable sensors could be used to passively collect physiological data related to burden symptoms. These very physiological cues could also be used to to determine the right time to trigger the conversation with the chatbot. Likewise, the user experience could be greatly enhanced by incorporating voice features instead of or in addition to text.

## Figures and Tables

**Figure 1 ijerph-19-16019-f001:**
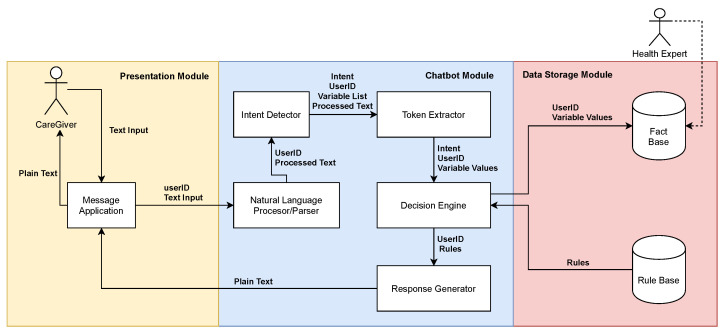
HIGEA architecture.

**Figure 2 ijerph-19-16019-f002:**
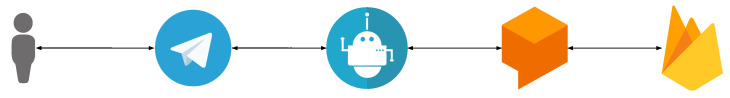
Communication flow chart of the HIGEA main components.

**Figure 3 ijerph-19-16019-f003:**
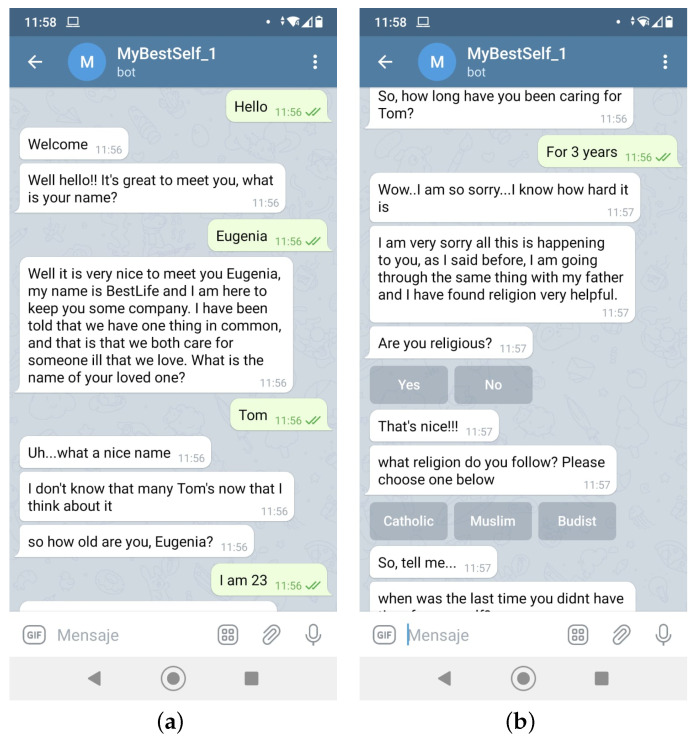
First conversation between the conversational agent and user. (**a**) At the beginning; (**b**) When the conversation progresses.

**Figure 4 ijerph-19-16019-f004:**
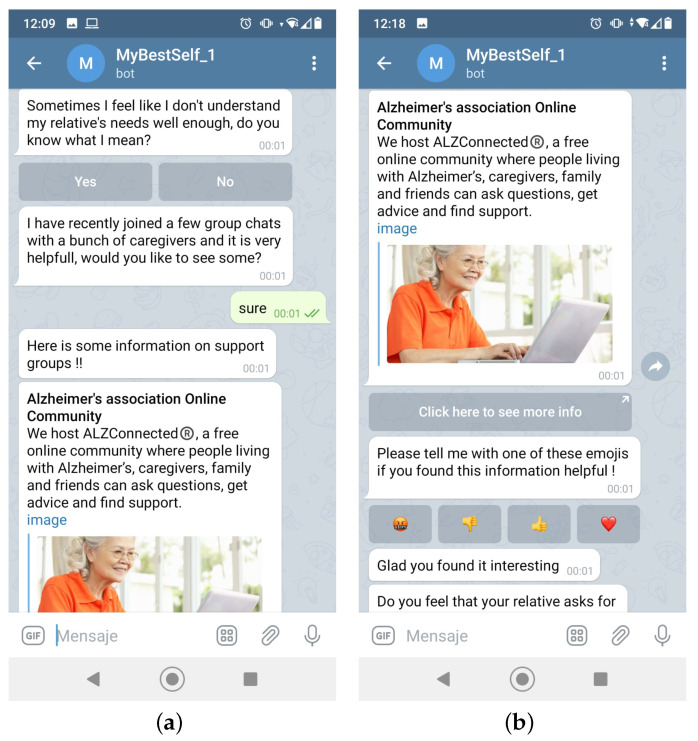
Small talk. (**a**) Example 1; (**b**) Example 2.

**Figure 5 ijerph-19-16019-f005:**
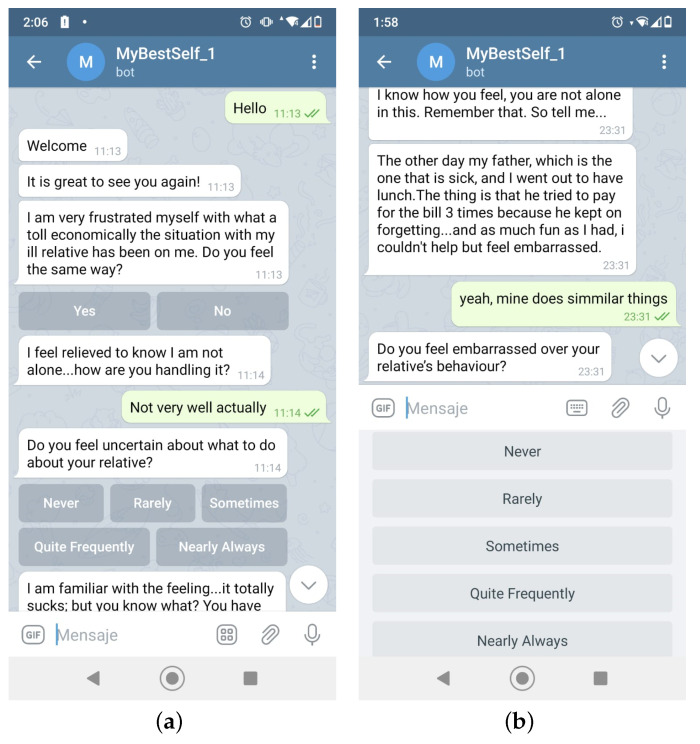
Zarit Test questions with different type of responses. (**a**) Responses as on-chat buttons; (**b**) Responses as inline keyboard buttons.

**Figure 6 ijerph-19-16019-f006:**
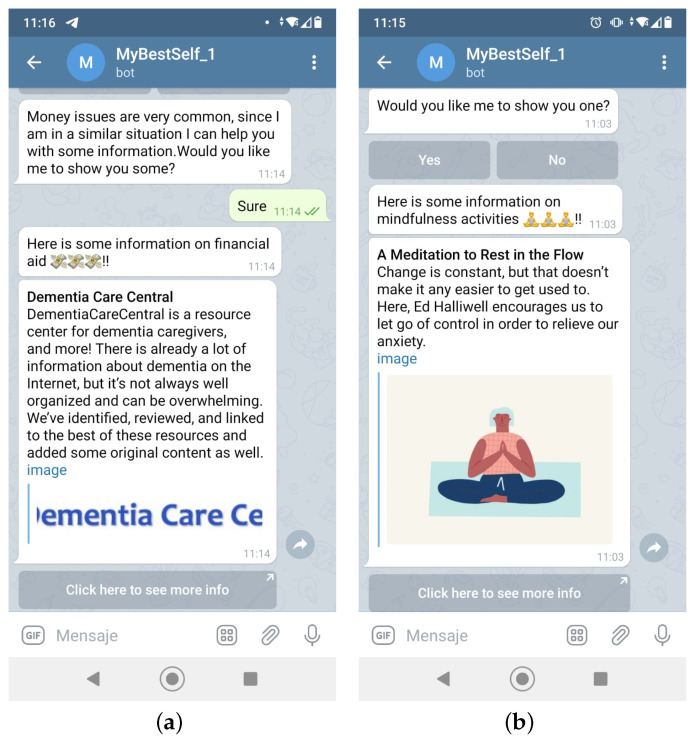
Examples of relevant information shared with the user. (**a**) Card with financial aid info; (**b**) Card with MBCT techniques.

**Figure 7 ijerph-19-16019-f007:**
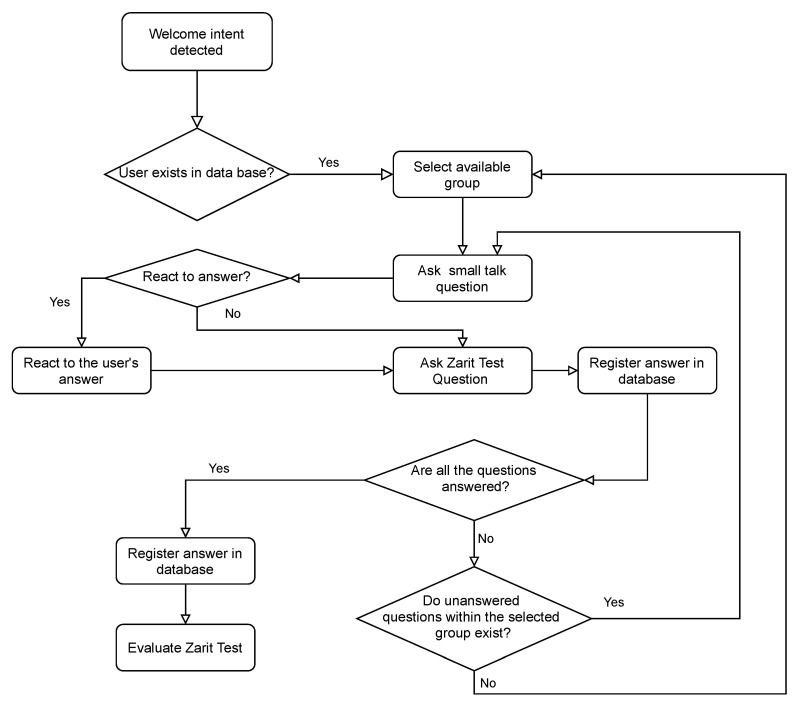
Conversation logic for a registered user.

**Figure 8 ijerph-19-16019-f008:**
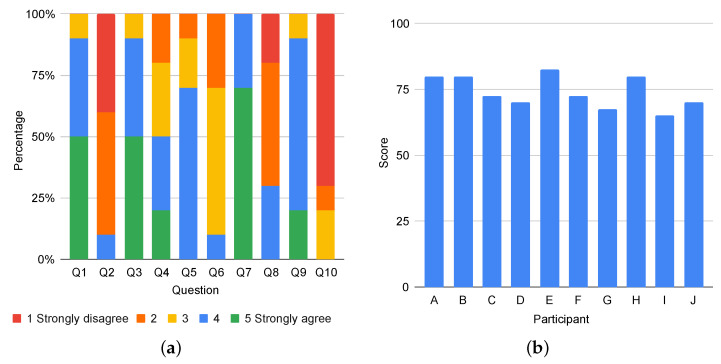
Results of the SUS questionnaire. (**a**) Distribution of the participants’ answers; (**b**) Overall score for each participant.

**Table 1 ijerph-19-16019-t001:** Zarit Test questions divided into conversational categories.

Group	A Social Life	B Money/Means	C Expectations	D Personal Values
**Question ID**	3, 6, 12, 13	7, 15, 16, 19	1, 8, 14, 18–22	2, 4, 5, 9, 10, 11, 17

**Table 2 ijerph-19-16019-t002:** Interpretation of score in the Zarit Burden Interview.

Score	Interpretation
0–20	Little to no burden
21–40	Mild to moderate burden
41–60	Moderate to severe burden
61–88	Severe burden

**Table 3 ijerph-19-16019-t003:** Options given to the user after evaluating the Zarit Test results.

Options	Score			
	0–20	21–40	41–60	61–88
Visualization	X	X		X
Breathing	X	X		X
Muscle	X	X		X
Testimonies		X	X	X
MBCT			X	X
Support			X	X
Economic			X	X
None	X	X	X	X

**Table 4 ijerph-19-16019-t004:** Demographics of the participants that used the HIGEA chatbot.

Participant	Gender	Age	Current Job	Sick Family Member Status
A	Female	53	Housekeeper	Father
B	Female	24	Civil Servant	Grandfather
C	Male	55	Civil Servant	Mother
D	Female	50	Lawyer	Partner
E	Female	15	Student	Father
F	Female	47	Baker	Father
G	Female	69	Housekeeper	Husband
H	Female	20	Student	Grandmother
I	Female	58	Health Professional	Great aunt
J	Male	59	Civil Servant	Mother

## Data Availability

All data collected in the usability study is provided directly in this manuscript.

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
