# Peer review of "HIGEA: An Intelligent Conversational Agent to Detect Caregiver Burden"

_ijerph, 2022, doi:10.3390/ijerph192316019_

Round 1
Reviewer 1 Report
The paper presents the development and test of an interesting tool with a clear objective.
The results seem to be promising. However, a few issues must be taken into account.
The advance over the state-of-the-art is not in the technological aspects, so this provides some margin for improvement. This includes use of AI techniques, voice instead or in addition to text, higher personalization, etc.
The evaluation has taken place over a rather small number of people. Although this has been cleary presented, this might have given partly wrong results and few specific advise for improvement.
The related work is in some cases rather old.
Author Response
Dear reviewer,
Thank you for your comments and suggestions. Please find a response to the points you raised in the cover letter in the attachment.
Sincerely,
Oresti Banos

Reviewer 2 Report
According to the title, this research aims at detecting the intensity of burdens on family caregivers. I unfortunately not able to figure out the objectives beyond detecting the intensity of burdens in caring for ill relatives. The authors should provide this. It will help to understand the study background.
The related works presented are a bit messy and neither narrowly handle similar prior works nor show limitation of the existing.
The study population is limited or poorly selected and described and show that the authors do not sufficiently do their preparation before starting.
- the study population does include working family caregivers students taking care of ill parents and siblings but does mention if the professional constraints conflict with caring for family ill person.
Beyond this, the study did not assess key factors impacting (increase g or decreasing) the intensity of burdens. The conversation snapshots confirms that.
I got the feeling (this is confirmed by the results as described) that the authors objectives are to digitize the Zarit questionnaires, which today are limited in some manners and need to be extended to better assess current factors amplifying or reducing the burdens intensities. Additionally, detailing the burdens parameters e.g., noctural stress/rest, sociall isolation, financial stress, will help to narrowly assess which parameters mostly put burdens on who ( I mean working, non-working and student caregivers).
System requirements/Arcitecture:
I got confused here. The title suggests to detect the burdens intensities, and here focused nod the deployment aspects of the system instead of on how the design should assist to attend this primary objectives.
I am wondering why AI is needed here to make decision since sarin questionnaires provide clearly score range for each question.
The authors need to explicitly describe what are the exact role (how to) of AI in assessing the score.
Results
Nothing to do with detecting burdens on caregivers. Instead it present or focused on the useability of the system.
Discussion
Unfortunately, the section does not discuss to what extent the system assesses the burden. Instead of this it is focused on the useability of the system. It is the objective of the study, does not it?
Recommendation:
https://www.researchgate.net/publication/333356311_IoT-Enabled_Health_Monitoring_and_Assistive_Systems_for_In_Place_Aging_Dementia_Patients_and_Elderly
Author Response

(The authors gave the same response as above.)

Round 2
Reviewer 2 Report
I went through your answers to my previous comments.
Thanks for your clarification.